# Favorable Effects of GLP-1 Receptor Agonist against Pancreatic β-Cell Glucose Toxicity and the Development of Arteriosclerosis: “The Earlier, the Better” in Therapy with Incretin-Based Medicine

**DOI:** 10.3390/ijms22157917

**Published:** 2021-07-24

**Authors:** Hideaki Kaneto, Tomohiko Kimura, Masashi Shimoda, Atsushi Obata, Junpei Sanada, Yoshiro Fushimi, Shuhei Nakanishi, Tomoatsu Mune, Kohei Kaku

**Affiliations:** 1Department of Diabetes, Endocrinology and Metabolism, Kawasaki Medical School, Kurashiki 701-0192, Japan; tomohiko@med.kawasaki-m.ac.jp (T.K.); masashi-s@med.kawasaki-m.ac.jp (M.S.); obata-tky@med.kawasaki-m.ac.jp (A.O.); gengorou@med.kawasaki-m.ac.jp (J.S.); fussy.k0113@med.kawasaki-m.ac.jp (Y.F.); nshuhei@med.kawasaki-m.ac.jp (S.N.); mune@med.kawasaki-m.ac.jp (T.M.); 2General Medical Center, Kawasaki Medical School, Kurashiki 701-0192, Japan; kka@med.kawasaki-m.ac.jp

**Keywords:** pancreatic β-cells, glucose toxicity, arteriosclerosis, GLP-1 receptor agonist, incretin-based medicine

## Abstract

Fundamental pancreatic β-cell function is to produce and secrete insulin in response to blood glucose levels. However, when β-cells are chronically exposed to hyperglycemia in type 2 diabetes mellitus (T2DM), insulin biosynthesis and secretion are decreased together with reduced expression of insulin transcription factors. Glucagon-like peptide-1 (GLP-1) plays a crucial role in pancreatic β-cells; GLP-1 binds to the GLP-1 receptor (GLP-1R) in the β-cell membrane and thereby enhances insulin secretion, suppresses apoptotic cell death and increase proliferation of β-cells. However, GLP-1R expression in β-cells is reduced under diabetic conditions and thus the GLP-1R activator (GLP-1RA) shows more favorable effects on β-cells at an early stage of T2DM compared to an advanced stage. On the other hand, it has been drawing much attention to the idea that GLP-1 signaling is important in arterial cells; GLP-1 increases nitric oxide, which leads to facilitation of vascular relaxation and suppression of arteriosclerosis. However, GLP-1R expression in arterial cells is also reduced under diabetic conditions and thus GLP-1RA shows more protective effects on arteriosclerosis at an early stage of T2DM. Furthermore, it has been reported recently that administration of GLP-1RA leads to the reduction of cardiovascular events in various large-scale clinical trials. Therefore, we think that it would be better to start GLP-1RA at an early stage of T2DM for the prevention of arteriosclerosis and protection of β-cells against glucose toxicity in routine medical care.

## 1. Introduction

The number of patients with type 2 diabetes mellitus (T2DM) has been increasing all over the world and T2DM is recognized as one of the most prevalent and serious metabolic diseases. In addition, economic and healthcare burden due to T2DM is a matter of concern at present. Therefore, it is very important to clarify the molecular mechanism for the pathophysiology of T2DM. Two main characteristics of T2DM are the pancreatic β-cell dysfunction and insulin resistance in various insulin target tissues such as the liver, skeletal muscle and adipose tissues. Normal β-cells can compensate for insulin resistance by increasing insulin secretion or β-cell mass, but insufficient compensation leads to the onset of T2DM. After then, once hyperglycemia becomes apparent, the β-cell function gradually deteriorates and insulin resistance aggravates.

It is well known that incretins, glucagon-like peptide-1 (GLP-1) and glucose-dependent insulinotropic polypeptide (GIP) have pleiotropic effects on a variety of tissues including pancreatic β-cells, artery, heart, liver, neuron and adipose tissues. While there are various GLP-1 target tissues, GLP-1 plays a crucial role in pancreatic β-cells; GLP-1 binds to the GLP-1 receptor (GLP-1R) in the β-cell membrane and thereby enhances insulin secretion, suppresses apoptotic cell death and increases the proliferation of β-cells. However, GLP-1R expression is reduced under diabetic conditions (Figure 1). The glucagon-like peptide-1 receptor agonist (GLP-1RA) and dipeptidyl peptidase-IV (DPP-IV) inhibitor are very often used in subjects with type 2 diabetes mellitus (T2DM). The DPP-IV inhibitor suppresses activity of DPP-IV, which is a splitting enzyme of incretin and increases serum levels of GLP-1 and GIP. Both incretins stimulate insulin secretion in a glucose-dependent manner and GLP-1 suppresses glucagon secretion, leading to the amelioration of glycemic control. Furthermore, the GLP-1RA and DPP-IV inhibitor do not usually bring about hypoglycemia and/or weight gain. GLP-1RA markedly increases circulating GLP-1 levels and functions at very high concentrations. Thereby, GLP-1RA has more potent glucose-lowering effects compared to the DPP-IV inhibitor. In addition, GLP-1RA reduces body weight by increasing central satiety and delaying gastric emptying. On the other hand, it has been drawing much attention to the idea that GLP-1 signaling is important in arteries as well. Indeed, it has been reported recently that administration of GLP-1RA leads to the reduction of cardiovascular events.

## 2. Incretin and Pancreatic β-Cells

T2DM is characterized by pancreatic β-cell dysfunction and insulin resistance. It has been shown that chronic hyperglycemia leads to the decrease of insulin biosynthesis and secretion due to β-cell glucose toxicity [1,2,3,4,5,6,7,8,9,10,11,12] and that the reduction of insulin mRNA expression is accompanied by decreased nuclear expression of insulin transcription factors such as MafA and PDX-1 [13,14,15,16,17,18,19,20,21]. Such phenomena are known as β-cell glucose toxicity. It has been shown, however, that β-cell function is recovered by the treatment with various antidiabetes medicine at an early stage of T2DM to some extent [22,23,24,25,26,27].

In response to the ingestion of food, GLP-1 and GIP are secreted from the gastrointestinal tract and stimulate insulin secretion from pancreatic β-cells. Both incretin hormones bind to each receptor in the β-cell membrane, which leads to enhancing insulin secretion, reducing β-cell apoptosis and promoting β-cell proliferation. Such an action of incretin hormones, however, is significantly reduced under diabetic conditions in humans and rodents such as mice and rats. It has been reported that expression levels of incretin receptors are reduced under diabetic conditions, which is probably involved in the impaired incretin effects and the development of β-cell failure found in T2DM [28,29,30]. For example, such downregulation of incretin receptor expression has been shown in obese type 2 diabetic mice (at 16 weeks of age, non-fasting blood glucose levels were about 500 mg/dL) and 90% partial pancreatectomized rats (4 weeks after the operation, fasting blood glucose levels were about 200 mg/dL). The precise mechanism for reduction of GLP-1 and GIP receptor levels under diabetic conditions remained unraveled. It has been shown recently, however, that the reduction of the transcription factor 7-like 2 (TCF7L2) expression level, which is a transcription factor and plays a crucial role in the maintenance of β-cell function, is involved in the downregulation of incretin receptor expression in β-cells [31,32,33]. It has been reported that TCF7L2 is involved in insulin biosynthesis, secretion and preservation of β-cell mass through the AKT and mTOR pathway [31,32,33,34,35,36,37]. Indeed, it is known that inactivation of TCF7L2 leads to the impairment of insulin secretion and glucose tolerance. Since TCF7L2 is a downstream factor transcription of the β-catenin signaling pathway, TCF7L2 is physiologically regulated by β-catenin. In addition, TCF7L2 is regulated by its genetic variation. Indeed, it is known that common genetic variations of TCF7L2 are associated with T2DM and that the subjects with the high-risk allele of TCF7L2 show impaired insulin secretion [38,39,40,41,42].

## 3. GLP-1-Stimulated Insulin Secretion

Insulin secretion is regulated by various intracellular signals in β-cells. Among them, cyclic adenosine monophosphate (cAMP) is particularly important for amplifying insulin secretion. While GLP-1RA and DPP-IV inhibitors have been often used in clinical practice, such incretin-based drugs function through the cAMP signaling. It is thought that cAMP potentiates insulin secretion through protein kinase A (PKA) phosphorylation of factors, which are associated with insulin secretory process. However, it has been proposed that there is another pathway for cAMP-induced insulin secretion; it was shown that cAMP had another target named Epac (also called as cAMP-GEF) in β-cells [43,44,45,46]. It is known that Epac signaling regulates cAMP-induced insulin granule exocytosis through the enlargement of the size of a readily released pool.

In addition, it was reported recently that a physiologically low concentration of GLP-1 activated protein kinase C (PKC) without a significant increase of intracellular cAMP, which also led to the enhancement of insulin secretion [47,48]. Thereby, it is likely that GLP-1 stimulates insulin secretion in a PKC-dependent or PKA-dependent manner, depending on its concentration. They showed that GLP-1 increased intracellular diacylglycerol and Ca^2+^ and activated PKC, leading to membrane depolarization and subsequent stimulation of insulin secretion. They also showed that the depolarizing effect of GLP-1 on electrical activity was mimicked by a PKC activator without activation of the PKA pathway. These new findings clearly indicate that circulating physiological concentration of GLP-1 directly stimulates insulin secretion from pancreatic β-cells.

## 4. GLP-1RA and Pancreatic β-Cells

Incretin-based medicine such as the GLP-1RA and DPP-IV inhibitor ameliorate glycemic control and mitigate the progression of β-cell dysfunction in human subjects and animal models. It has been reported that GLP-1RA preserves pancreatic β-cells in various types of T2DM rodents [49,50,51,52,53,54,55]. For example, it was shown that when T2DM db/db mice at 10 weeks old were treated with GLP-1RA (liraglutide) for 2 weeks, metabolic variables and insulin sensitivity were improved. GLP-1RA also increased glucose-stimulated insulin secretion (GSIS) and islet insulin content and reduced triglyceride content in islets. Furthermore, expression levels of various genes related to proapoptosis, ER stress and lipid synthesis were downregulated whereas those related to antiapoptosis and antioxidative stress were upregulated. GLP-1RA treatment for 2 days slightly improved metabolic variables in db/db mice, but GSIS, insulin and triglyceride content were not affected. Such treatment increased gene expression related to cell differentiation, proliferation and antiapoptosis and suppressed gene expression involved in proapoptosis, although there was no effect on oxidative stress- or ER stress-related factors [49]. Taken together, GLP-1RA increases β-cell mass not only by directly regulating cell kinetics, but also by suppressing oxidative and ER stress, secondary to the amelioration of glucolipotoxicity.

Protective effects of GLP-1RA on β-cells were reported in another type of diabetic mice [52,53,54,55]. For example, it was shown that GLP-1RA improved pancreatic β-cell mass and function in alloxan-induced diabetic mice [52]. They examined the effects of GLP-1RA on β-cell fate and function by using an inducible Cre/loxP system. In the results, chronic GLP-1RA treatment for 30 days improved glucose tolerance and insulin response to oral glucose load. Additionally, GLP-1RA treatment doubled β-cell mass compared to the vehicle group by increasing the β-cell proliferation rate and reducing apoptotic cell death. Interestingly, however, there was no or little contribution of neogenesis to such an increase in β-cell mass based on the data obtained with the Cre/loxP system. In addition, GLP-1RA reduced oxidative stress in pancreatic islets. Furthermore, the beneficial effects of GLP-1RA in these mice were maintained 2 weeks after drug withdrawal [52]. In another study, it was shown in more detail how GLP-1RA preserved β-cell mass [54]. They showed that GLP-1RA protected mouse pancreatic β-cell line βTC6 cells from serum withdrawal-induced apoptosis through the inactivation of caspase-3. They also showed that PI3-kinase-dependent AKT phosphorylation, inactivation of the proapoptotic protein BAD and inhibition of the FoxO1 transcription factor were involved in antiapoptotic action of GLP-1RA [54].

It has been reported that GLP-1RA shows more favorable effects at an early stage of T2DM compared to an advanced stage [50,51]. T2DM db/db mice were treated with GLP-1RA (liraglutide) and/or pioglitazone for 2 weeks at an early (7 weeks old) and advanced stage (16 weeks old). At an early stage, insulin biosynthesis and secretion were markedly increased by such a treatment, which was not observed at an advanced stage. Expression levels of various β-cell-related factors such as MafA and PDX-1 were upregulated by such a treatment only at an early stage. It is likely that the recovery of MafA expression after such treatment is particularly important for the recovery of the β-cell function and amelioration of glycemic control, because MafA regulates not only insulin but also various factors related to GSIS. The increased expression of GLUT2 and glucokinase could also explain the augmentation of GSIS observed at an early stage. In addition, the expression level of GLP-1R was reduced at an advanced stage, which we think explains the reason why GLP-1RA showed more favorable effects at an early stage. Furthermore, β-cell mass and proliferation were increased by the treatment only at an early stage. [51]. Similarly, it was shown that DPP-IV inhibitor together with the SGLT2 inhibitor exerted more favorable effects on the β-cell function and mass at an early stage of T2DM compared to an advanced stage [56]. It is well known that GLP-1 binds to its receptor in the β-cell membrane and activates adenylate cyclase and the cAMP/PKA signaling pathway [43,44,45,46]. In addition, a low concentration of GLP-1 activates PKC without a significant increase of intracellular cAMP, which also leads to the enhancement of insulin secretion [47,48]. It is known that the activation of such kinases is involved in the reduction of β-cell apoptosis, proliferation of β-cells and maintenance of β-cell function. In this study, expression of the GLP-1R level was downregulated at an advanced stage compared with an early stage. Therefore, we think that the downregulation of signal pathways such as PKA or PKC is, at least in part, involved in the ineffectiveness of GLP-1RA on the β-cell function at an advanced stage. Taken together, the usage of incretin-based medicine at an early stage of T2DM would be useful and promising for the preservation of the β-cell function and mass.

On the other hands, it is well known that chronic exposure to a large amount of ligand leads to the downregulation of its receptor. In addition, it is known that the serum GLP-1 level becomes very high after the usage of GLP-1RA, which is a ligand of GLP-1R. It remained unknown, however, whether the long-time usage of GLP-1RA downregulates its receptor. It was reported that GLP-1R expression was reduced after long-term exposure to GLP-1RA (dulaglutide) in non-diabetic and diabetic mice. Obese type 2 diabetic db/db mice and non-diabetic db/m mice were treated with GLP-1RA or the control vehicle for 17 weeks (from 7 to 24 weeks of age). Various metabolic parameters such as GSIS, insulin and triglyceride content in islets and β-cell-related gene expression were evaluated after the intervention. In db/m mice, GLP-1R expression in β-cells was not decreased even after long-term administration of GLP-1RA. In db/db mice, GLP-1R expression at 24 weeks of age was significantly lower compared to that at 7 weeks probably due to glucose toxicity [57]. Furthermore, GLP-1R expression in 24-week-old db/db mice treated with GLP-1RA was higher, rather than downregulated, compared to 24-week-old untreated diabetic mice, which was probably due to the amelioration of glycemic control. Food intake and blood glucose levels in db/db mice treated with GLP-1RA were lower until 24 weeks of age compared to untreated db/db mice. Expression levels of various β-cell-related genes, insulin biosynthesis and secretion were enhanced after GLP-1RA treatment in db/db mice. In contrast, oxidative and endoplasmic reticulum stress, inflammation, fibrosis and apoptosis were suppressed after GLP-1RA treatment [57]. Taken together, GLP-1RA shows favorable effects on glycemic control and protective effects on pancreatic β-cells for a long period without reducing the GLP-1R expression level.

GLP-1R is one of the G protein-coupled receptors (GPCRs). GLP-1 binds to its receptor on the cell membrane and the complex of GLP-1 and its receptor GPCR is internalized into cells. In general, it is thought that the internalized receptor preserves its expression level compared to a non-internalized receptor. Consequently, we think that such characteristics of GLP-1R could explain the reason why GLP-1R expression was not decreased even after long-term administration of GLP-1RA. In addition, some drug therapy has been developed by utilizing the phenomena that chronic exposure to a large amount of ligand downregulates its receptor expression. For example, in the treatment for endometriosis, the gonadotropin releasing hormone (GnRH) agonist suppresses the production of the downstream hormone through downregulation of its receptor by continuous administration of a large amount of ligand [58,59,60,61].

## 5. GLP-1RA and Arteriosclerosis

### 5.1. Incretin Signaling and Arterial Cells

GLP-1R expression is observed in endothelial and smooth muscle cells. In endothelial cells, incretin signaling improves the vascular relaxation response through eNOS expression and activity and retards the development of arteriosclerosis (Figure 2) [62,63,64]. Activation of GLP-1 signaling in arteries leads to the mitigation of inflammatory cytokines. In arterial cells, GLP-1 signaling improves the wall disorder induced by various factors including hyperglycemia and inflammatory cytokines. In vascular smooth muscle cells, GLP-1R stimulation prevents the development of arteriosclerosis. In addition, although GLP-1R is expressed in various cell types, it was not clearly elucidated how GLP-1RA can retard the progression of arteriosclerosis. However, recently the vasoprotective mechanism of GLP-1RA was clearly demonstrated at the cellular level by using global GLP-1R knockout mice, endothelial cell-specific GLP-1 knockout mice and myeloid cell-specific GLP-1R knockout mice. As the results, it was shown that GLP-1RA treatment normalized blood pressure, cardiac hypertrophy, vascular fibrosis, endothelial dysfunction, oxidative stress and vascular inflammation in an endothelial GLP-1R-dependent manner [65]. We think that these novel findings are strong evidence showing that endothelial GLP-1R expression is critical for GLP-1 to fully show their effects in arteries. Incretin-based therapy substantially ameliorates glycemic control without hypoglycemia and/or weight gain, which leads one to the prevention of diabetic macroangiopathy. In addition, GLP-1 has direct protective effects on vascular cells via GLP-1R. Therefore, it is likely that incretin-based therapy shows favorable effects on the development of arteriosclerosis through the reduction of blood glucose levels and their direct effects on arterial cells via GLP-1R.

### 5.2. Downregulation of GLP-1R Expression in Arterial Cells under Diabetic Conditions

GLP-1R expression in pancreatic β-cells is reduced under diabetic conditions and TCF7L2 is known to function as a transcription factor for GLP-1R at least in β-cells. Incretin signaling is known to prevent the development of arteriosclerosis by the relaxation response in endothelial cells via the GLP-1R. It was reported recently that GLP-1R and TCF7L2 expression levels in endothelial and smooth muscle cells were significantly lower in obese type 2 diabetic db/db mice compared to non-diabetic db/m mice [66]. Furthermore, reduction of the TCF7L2 level using siTCF7L2 resulted in the downregulation of GLP-1R expression in cultured vascular endothelial cells. In addition, when the TCF7L2 level was enhanced using the TCF7L2 expressing adenovirus, the GLP-1R expression level was substantially augmented [67]. In conclusion, the GLP-1R expression level was substantially reduced under diabetic conditions together with the decrease of the TCF7L2 level (Figure 2). Furthermore, it was shown that the TCF7L2 is a possible regulator of the GLP-1R expression in the artery as reported in β-cells.

### 5.3. Favorable Antiarteriosclerotic Effects of GLP-1RA

It was not known so far whether or not there was some difference in effectiveness of GLP-1RA between an early and an advanced stage of T2DM. Recently, however, to address such questions, either GLP-1RA (dulaglutide) or the vehicle was administered to streptozotocin-induced diabetic ApoE knockout mice from 10 to 18 weeks of age as an early stage and from 18 to 26 weeks as an advanced stage. In the results, in an early stage group, the arteriosclerotic lesion in the aortic arch and Mac-2 and CD68-positive areas in the aortic root were significantly smaller in the GLP-1RA group [68]. In the abdominal aorta, expression levels of various inflammation markers were lower in the GLP-1RA group. In an advanced stage group, there were no immunohistological differences in the aortic root and expression levels of various factors between the GLP-1RA and vehicle group [68]. Taken together, GLP-1RA shows more favorable antiarteriosclerotic effects at an early stage of T2DM compared to an advanced stage.

### 5.4. Protective Role of GLP-1RA against Cardiovascular Events in Subjects with T2DM

While cardiovascular events sometimes bring about serious and lethal situations, it has been shown recently that GLP-1RAs reduce cardiovascular events [69,70,71,72,73,74]. The LEADER trial showed the effects of a once-daily injection of GLP-1RA liraglutide on cardiovascular events. The primary composite cardiovascular outcome was observed in significantly fewer patients in the treatment group (hazard ratio (HR): 0.87). Additionally, fewer patients died from cardiovascular causes in the treatment group (HR: 0.78). The rate of the all-cause death was also lower in the treatment group (HR: 0.85) [69,70]. The REWIND trial showed the effects of a once-weekly injection of dulaglutide on cardiovascular events. HR in the primary composite cardiovascular outcome was 0.88 and that in the all-cause death was 0.90 [71,72]. The SUSTAIN-6 trial showed the effects of a once-weekly injection of semaglutide on cardiovascular events. The occurrence of the primary composite cardiovascular outcome was lower in the treatment group (HR: 0.74). HR in nonfatal myocardial infarction and nonfatal stroke was 0.74 and 0.61, respectively [73,74]. Taken together, a once-daily injection of liraglutide and once-weekly injection of dulaglutide and semaglutide are expected to prevent major adverse cardiovascular events. The above-mentioned three large-scale clinical trials strongly support the idea that GLP-1RAs show a protective role against cardiovascular events in subjects with T2DM. Therefore, in routine medical care we should willingly use GLP-1RA in subjects with T2DM especially in subjects with a large risk of cardiovascular events.

## 6. Conclusions

In this review article, we featured roles of GLP-1 signaling in pancreatic β-cells and arteries. In addition, we described the usability of GLP-1RA based on the molecular mechanism for β-cell glucose toxicity and for the development of arteriosclerosis.

Our current ideas about pancreatic β-cell glucose toxicity are as follows. First, chronic hyperglycemia leads to a decrease of insulin biosynthesis and/or secretion in the diabetic state, which is accompanied by decreased expression of insulin transcription factors. In routine medical care, it is very important to alleviate such β-cell glucose toxicity in order to prevent the aggravation of T2DM. Second, incretin signaling plays crucial roles in pancreatic β-cells and we believe that GLP-1RA is a promising medicine to protect β-cells against glucose toxicity. However, incretin sensitivity in β-cells is weakened under diabetic conditions, at least in part, due to downregulation of GLP-1R expression, which we think may be associated with the aggravation of β-cell glucose toxicity.

Our current ideas about the usability of GLP-1RA are as follows. First, as described above, GLP-1R expression in pancreatic β-cells is downregulated under diabetic conditions. The data also suggest that it would be better to use incretin-based medicine at an early stage of T2DM. Indeed, GLP-1RA showed more favorable effects on β-cells at an early stage in T2DM mice. Second, incretin signaling plays a crucial role against arteriosclerosis, but incretin sensitivity in arteries is also weakened, at least partially, due to downregulation of GLP-1R expression, which we think may facilitate the development of arteriosclerosis. Consequently, we think that it would be better to use incretin-based medicine at an early stage of T2DM for the prevention of arteriosclerosis. Indeed, GLP-1RA showed more favorable effects against the progression of arteriosclerosis at an early stage. Third, a series of large-scale clinical trials have shown that GLP-1RAs have favorable effects against the onset of cardiovascular events. Taken together, incretin signaling plays crucial roles in various kinds of cells such as pancreatic β-cells and arterial cells and GLP-1RAs are promising from the clinical points of view and basic research area.

## Figures and Tables

**Figure 1 ijms-22-07917-f001:**
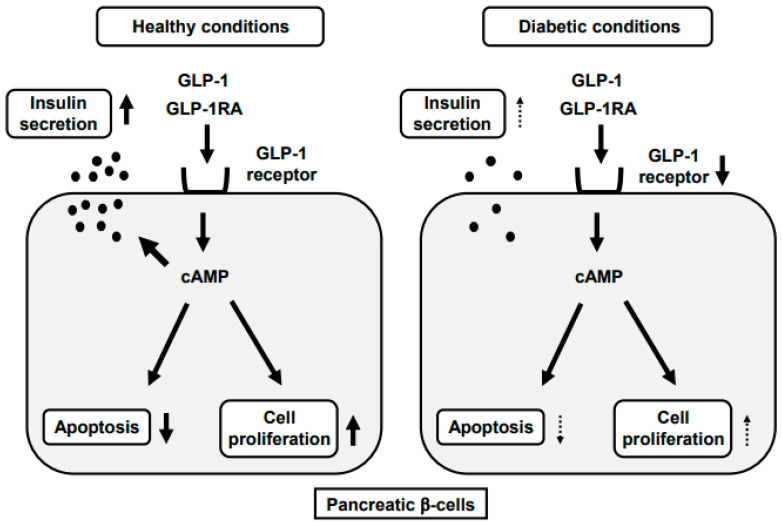
Reduction of GLP-1RA effects on pancreatic β-cells under diabetic conditions. GLP-1 binds to its receptor in pancreatic β-cells, which leads to the enhancement of insulin secretion, reduction of apoptotic cell death and increase of β-cell proliferation. After chronic exposure to hyperglycemia, however, GLP-1 receptor expression in β-cells is reduced, which weakens the protective effects of GLP-1 and GLP-1RA against β-cells glucose toxicity.

**Figure 2 ijms-22-07917-f002:**
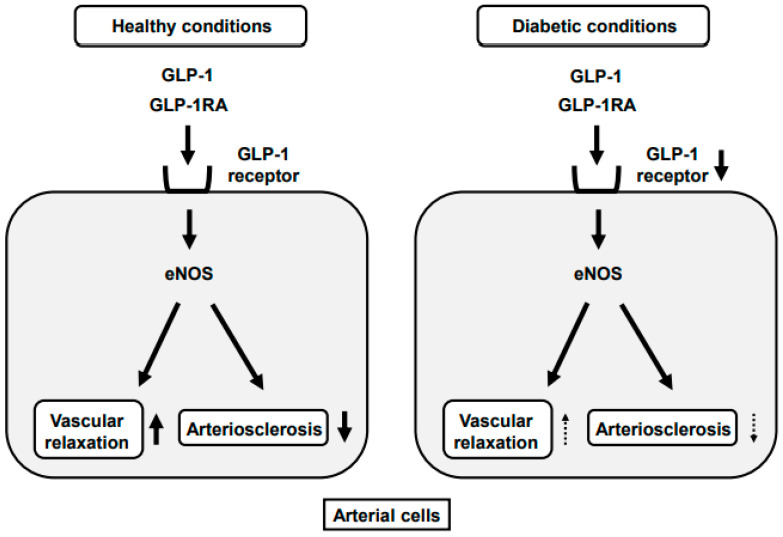
Reduction of GLP-1RA effects on arterial cells under diabetic conditions. GLP-1 binds to its receptor in arterial cells, which leads to the enhancement of vascular relaxation and the prevention of arteriosclerosis. After chronic exposure to hyperglycemia, however, GLP-1 receptor expression in arterial cells is reduced, which weakens the protective effects GLP-1 and GLP-1RA against arteriosclerosis.

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
