# Peer review of "Favorable Effects of GLP-1 Receptor Agonist against Pancreatic β-Cell Glucose Toxicity and the Development of Arteriosclerosis: “The Earlier, the Better” in Therapy with Incretin-Based Medicine"

_ijms, 2021, doi:10.3390/ijms22157917_

Round 1

Reviewer 1 Report

In the present manuscript entitled "Favorable effects of GLP-1 receptor activator against pancreatic b-cell glucose toxicity and the development of arteriosclerosis: "The earlier, the better" in therapy with incretin-based medicine. The authors attempt to review the current state of the art of GLP1-RA on pancreatic beta function associated with glucotoxicity and the development of arteriosclerosis. This is a relevant topic, and to bring together information on an intense work since there is a lot of information on the subject and these GLIP-1 analogs are currently being tested as drugs in type 2 diabetes.

Although the information shown is relevant, the present form of the paper has multiple shortcomings that make it unsuitable for publication in this version and should be improved.

The most relevant criticisms that this referee considers for his decision are:

-The first time the acronym GIP appears is in the introduction, but the authors define the acronym in section 3.

-In paragraph 3 the authors explain that "GLP-1 and glucose-dependent insulinotropic polypeptide (GIP) are secreted from the gastrointestinal tract and facilitate insulin secretion from pancreatic cells". In the opinion of this referee, this is an overly simplistic explanation, it cannot be said that incretins facilitate insulin release. In any case, it would be a stimulation of insulin secretion. Facilitating is not an appropriate term.

-In the same way, as noted above, the authors point out in section 3 that "Both incretin hormones bind to each receptor in b-cell membrane, which leads to enhance insulin secretion, reduce b-cell apoptosis and facilitate b-cell proliferation." The term facilitate is not correct, in any case, we would speak of promoting proliferation and therefore the survival of beta cells.

-The authors devote a section to the transcription factor TCF7L2. However, the physiological role of this transcription factor in the maintenance of beta-cell function is not clear. The authors only point out that it has a crucial role in the maintenance of β-cell function. Could the authors elaborate on the explanation? What is this role?

How is it regulated physiologically?

-"Decreased expression of TCF7L2 and incretin receptors has been demonstrated in cells under diabetic conditions, both in humans and in rodents such as mice and rats." Could the authors elaborate on the conditions under which the expression of this factor is reduced?

-The acronym GLP-1RA is already introduced in the first section, so it is redundant that its definition appears again in section 5.

-It would be convenient to expand sections 4 and 5 and include figures.

-In section 5, the authors explain only one study of their own, which, although relevant, is old. The authors do not compare or discuss what other authors and experts in this field have contributed. From 2011 to the present, there is a multitude of studies in this regard, which, in the opinion of this referee, should be included to give real information on the state of the art of effect of liraglutide on pancreatic beta function. (Some examples of relevant bibliography to discuss in this section 25938469, 31412267, 30177467, 25710926, 29524296).

-Same criticism for points 6 and 7.

-In general, in the whole section that would include glucotoxicity and diabetes (1-7) the authors should include more studies. They mostly focus on their own findings, which decontextualizes the review.

-It is recommended to improve the explanation of GLP-1 receptor activators. For example, the authors begin to talk about dulaglutide after an extensive section on liraglutide. A better explanation of these analogs, their differences, and other studies on them should be included. Semaglutide is discussed later in section 12. Comprehension is chaotic both because of the order and the lack of information.

-It is suggested to include some explanatory figures to improve reading. The review only includes one figure, which makes it difficult to understand the review.

-From the point of view of this referee, the present review is very simplistic and descriptive; the authors limit themselves in most cases to listing studies and their main conclusions, without delving into mechanisms or explanations and without a critical discussion of them.

-Section 2 of the review is out of focus. This referee does not understand this part which is not discussed again in the whole work. The authors should focus more on what they are trying to convey in this section and its relevance to subsequent sections.

-The titles of the different sections should be more concise, as they are sometimes highlighted in bold. This makes readers lose interest and the reading becomes dense. From the point of view of this referee, the present review is difficult to follow and dense, and more explanatory figures should be included to help navigate the text. The authors should include more studies by other authors with which to compare their own.

Author Response

Thank you very much for your valuable suggestion. According to your kind suggestion, we responded to your comments and amended the manuscript.

Reviewer 2 Report

The authors focused on the pancreatic β-cell and artery, and described clinical perspectives of GLP-1 receptor agonists on their functions in patients with type 2 diabetes. This review contains the latest reports and is informative for researchers and physicians. However, the reviewer raised several minor comments:

  1. The authors defined GLP-1RA as “a GLP-1 receptor activator”. For instance, liraglutide is recognized as “a GLP-1 receptor agonist”. Please reconsider the definition.
  2. The flow of abstract is not smooth due to a duplicate of sentence and grammatical errors.
  3. In the Introduction, at first, the authors should briefly describe pathophysiology of T2DM, and fundamental findings and characteristics of GLP-1 and GLP-1R. And then the authors should explain systemic and local effects of DPP-4 and incretin. In addition, GLP-1R is present not only in β-cells but also in the neuron, adipose tissue, liver, heart and so forth. Thus, the authors should clarify their focus of this paper.
  4. What is insulin gene (page 4)? Does it mean gene involved in insulin secretion?
  5. In page 4, the authors explained a roll of MafA only in knockout and overexpression animals. How about diabetic patients or animals? Does MafA expression affect onset of diabetes?
  6. Considering the title of this paper, the structure of page 4 is not reasonable. Please explain the relationship among GLP-1RA, PDX-1, and MafA.
  7. Fine sectioning will make the readers difficult to get the whole story of this paper. Please reconsider the entire structure. At least 2-3, 5-7, and 8-11 could be integrated, respectively.
  8.  The authors introduced several molecules. To lead the readers to understand, the authors should make a table or a figure which summarize their functions and relationship between the molecules and glucose metabolism.

Author Response

(The authors gave the same response as above.)

Round 2

Reviewer 1 Report

In the final version, the figures do not appear, although the legends do. Perhaps there was a problem when uploading the final file. It is recommended to review it.

Author Response

Thank you very much for your kind message.

We carefully re-uploaded the text together with Figures through the online submission system.

Sincerely,

Hideaki  Kaneto